# Surgical Management of the Axilla in Clinically Node-Positive Breast Cancer Patients Converting to Clinical Node Negativity through Neoadjuvant Chemotherapy: Current Status, Knowledge Gaps, and Rationale for the EUBREAST-03 AXSANA Study

**DOI:** 10.3390/cancers13071565

**Published:** 2021-03-29

**Authors:** Maggie Banys-Paluchowski, Maria Luisa Gasparri, Jana de Boniface, Oreste Gentilini, Elmar Stickeler, Steffi Hartmann, Marc Thill, Isabel T. Rubio, Rosa Di Micco, Eduard-Alexandru Bonci, Laura Niinikoski, Michalis Kontos, Guldeniz Karadeniz Cakmak, Michael Hauptmann, Florentia Peintinger, David Pinto, Zoltan Matrai, Dawid Murawa, Geeta Kadayaprath, Lukas Dostalek, Helidon Nina, Petr Krivorotko, Jean-Marc Classe, Ellen Schlichting, Matilda Appelgren, Peter Paluchowski, Christine Solbach, Jens-Uwe Blohmer, Thorsten Kühn

**Affiliations:** 1Department of Obstetrics and Gynecology, Campus Lübeck, University Hospital of Schleswig Holstein, 23538 Lübeck, Germany; 2Medical Faculty, Heinrich-Heine-University Düsseldorf, 40225 Düsseldorf, Germany; 3Department of Gynecology and Obstetrics, Ente Ospedaliero Cantonale, Ospedale Regionale di Lugano, 6900 Lugano, Switzerland; MariaLuisa.Gasparri@eoc.ch; 4Faculty of Biomedicine, University of the Italian Switzerland (USI), 6900 Lugano, Switzerland; 5Department of Molecular Medicine and Surgery, Karolinska Institutet, 171 77 Stockholm, Sweden; jana.de-boniface@ki.se (J.d.B.); matilda.appelgren@ki.se (M.A.); 6Department of Surgery, Capio St. Göran’s Hospital, 112 19 Stockholm, Sweden; 7Breast Surgery Unit, San Raffaele Hospital Milan, 20132 Milano MI, Italy; gentilini.oreste@hsr.it (O.G.); dimicco.rosa@hsr.it (R.D.M.); 8Department of Gynecology and Obstetrics, University Hospital Aachen, 52074 Aachen, Germany; estickeler@ukaachen.de; 9Department of Gynecology and Obstetrics, University Hospital Rostock, 18059 Rostock, Germany; steffi.hartmann@kliniksued-rostock.de; 10Department of Gynecology and Gynecological Oncology, AGAPLESION Markus Krankenhaus, 60431 Frankfurt am Main, Germany; marc.thill@fdk.info; 11Breast Surgical Unit, Clínica Universidad de Navarra, 28027 Madrid, Spain; irubior@unav.es; 12Department of Surgical Oncology, “Prof. Dr. Ion Chiricuță” Institute of Oncology, 400015 Cluj-Napoca, Romania; bonci.eduard@gmail.com; 1311th Department of Oncological Surgery and Gynecological Oncology, “Iuliu Hațieganu” University of Medicine and Pharmacy, 400012 Cluj-Napoca, Romania; 14Breast Surgery Unit, Comprehensive Cancer Center, Helsinki University Hospital, University of Helsinki, 00280 Helsinki, Finland; laura.niinikoski@hus.fi; 151st Department of Surgery, Laiko Hospital, National and Kapodistrian University of Athens, 115 27 Athens, Greece; michalis_kontos@yahoo.com; 16Breast and Endocrine Unit, General Surgery Department, Zonguldak BEUN The School of Medicine, Kozlu/Zonguldak 67600, Turkey; gkkaradeniz@yahoo.com; 17Brandenburg Medical School Theodor Fontane, 16816 Neuruppin, Germany; Michael.Hauptmann@mhb-fontane.de; 18Institut für Pathologie, Medical University of Graz, 8010 Graz, Austria; florentia.peintinger@medunigraz.at; 19Champalimaud Clinical Center, Breast Unit, Champalimaud Foundation, 1400-038 Lisboa, Portugal; davidgomespinto@gmail.com; 20Department of Breast and Sarcoma Surgery, National Institute of Oncology, 1122 Budapest, Hungary; matraidoc@gmail.com; 21Collegium Medicum, University of Zielona Góra, 65-046 Zielona Góra, Poland; dmurawa@gmail.com; 22Breast Surgical Oncology and Oncoplastic Surgery, Max Institute of Cancer Care, Max Healthcare Delhi, Delhi 110092, India; dr_kaygee@hotmail.com; 23Gynecologic Oncology Center, Department of Obstetrics and Gynecology, First Faculty of Medicine, Charles University, General University Hospital, 128 00 Prague, Czech Republic; Lukas.Dostalek@vfn.cz; 24Oncology Hospital, University Hospital Center “Nene Tereza”, 1000 Tirana, Albania; ninahelios@yahoo.com; 25Petrov Research Institute of Oncology, 197758 Saint-Petersburg, Russia; dr.krivorotko@mail.ru; 26Department of surgical oncology, Institut de cancerologie de l’Ouest Nantes, 44800 Saint Herblain, France; jean-marc.classe@ico.unicancer.fr; 27Department for Breast and Endocrine Surgery, Oslo University Hospital, 0188 Oslo, Norway; elschl@ous-hf.no; 28Department of Gynecology and Obstetrics, Regio Klinikum Pinneberg, 25421 Pinneberg, Germany; peterpaluchowski@gmail.com; 29Breast Center, Department of Gynecology and Obstetrics, University of Frankfurt, 60590 Frankfurt am Main, Germany; Christine.Solbach@kgu.de; 30Department of Gynecology and Breast Cancer Center, Charite Berlin, 10117 Berlin, Germany; jens.blohmer@charite.de; 31Department of Gynecology and Obstetrics, Klinikum Esslingen, 73730 Esslingen, Germany; kuehn.thorsten@t-online.de

**Keywords:** neoadjuvant therapy, breast cancer, therapy response, targeted axillary dissection, marked lymph node

## Abstract

**Simple Summary:**

Currently, it is unclear which kind of axillary staging surgery breast cancer patients with lymph node metastasis should receive after neoadjuvant chemotherapy. For decades, these patients have been treated with a full axillary lymph node dissection, even if they converted to clinical node negativity. However, the removal of a large number of lymph nodes during the procedure can increase arm morbidity and impact quality of life. Therefore, several studies investigated less radical surgical strategies in this setting, such as sentinel lymph node biopsy or targeted axillary dissection, i.e., removal of a previously marked node combined with sentinel node removal. In this review, we summarize current evidence on the different surgical techniques and compare national and international recommendations. We show that many questions regarding oncological safety of different surgery types and the optimal marking technique remain unanswered and present the multinational prospective cohort study AXSANA that will address these open issues.

**Abstract:**

In the last two decades, surgical methods for axillary staging in breast cancer patients have become less extensive, and full axillary lymph node dissection (ALND) is confined to selected patients. In initially node-positive patients undergoing neoadjuvant chemotherapy, however, the optimal management remains unclear. Current guidelines vary widely, endorsing different strategies. We performed a literature review on axillary staging strategies and their place in international recommendations. This overview defines knowledge gaps associated with specific procedures, summarizes currently ongoing clinical trials that address these unsolved issues, and provides the rationale for further research. While some guidelines have already implemented surgical de-escalation, replacing ALND with, e.g., sentinel lymph node biopsy (SLNB) or targeted axillary dissection (TAD) in cN+ patients converting to clinical node negativity, others recommend ALND. Numerous techniques are in use for tagging lymph node metastasis, but many questions regarding the marking technique, i.e., the optimal time for marker placement and the number of marked nodes, remain unanswered. The optimal number of SLNs to be excised also remains a matter of debate. Data on oncological safety and quality of life following different staging procedures are lacking. These results provide the rationale for the multinational prospective cohort study AXSANA initiated by EUBREAST, which started enrollment in June 2020 and aims at recruiting 3000 patients in 20 countries (NCT04373655; Funded by AGO-B, Claudia von Schilling Foundation for Breast Cancer Research, AWOgyn, EndoMag, Mammotome, and MeritMedical).

## 1. Introduction

In breast cancer patients, the optimal surgical management of the axilla has been controversially discussed over the last two decades. For many years, axillary lymph node dissection (ALND) has been considered as the standard of care. The rationale behind this was firstly to assess the pathological lymph node status (diagnostic value, “staging”) and secondly to provide locoregional control (therapeutic value). Due to its high morbidity, this approach has gradually been abandoned in favor of sentinel lymph node biopsy (SLNB), a less invasive procedure for axillary staging of clinically node-negative patients, over the past two decades. In recent years, SLNB has also become standard of care for patients with clinically unsuspicious nodes at time of diagnosis who have completed neoadjuvant chemotherapy. The detection rate and accuracy of SLNB are excellent in this setting, and axillary recurrence rates are negligible [1,2,3].

Nonetheless, in patients with clinically apparent axillary lymph node metastases (cN+) at time of diagnosis who achieve complete clinical response in the axilla (ycN0) after neoadjuvant chemotherapy (NACT), it is unclear which axillary surgical staging strategy should be offered. This uncertainty is expressed in the heterogeneity of recommendations endorsed by different national and international societies, which range from SLNB to targeted axillary dissection (TAD) or ALND (Table 1 and Table 2). Some societies do not consider SLNB standard of care in this setting because of the relatively high false negative rates (FNRs) reported in large prospective trials and confirmed in a meta-analysis [1,4,5,6]. In the SENTINA and the ACOSOG Z1071 trials, FNRs were 12% and 14% respectively, and thus higher than the arbitrarily chosen but widely accepted cut-off value of 10% (Table 3). So far, however, only limited data regarding the oncologic outcome following SLNB alone are available in this setting [3].

Marking positive lymph nodes at the time of diagnosis and prior to neoadjuvant chemotherapy with clips, coils, radioactive seeds, or other markers may improve the FNR of de-escalated surgical staging procedures [7,8]. The best marking technique, however, has not been unanimously identified yet. Importantly, data comparing recurrence rates and surgical morbidity among SLNB, ALND, and TAD are so far not available.

As a result, the German S3 guidelines updated in 2020 recommend ALND after NACT in cN+ patients, as do Austrian and Scandinavian guidelines. As a contrast, high-impact networks such as European Society of Medical Oncology (ESMO) in Europe and National Comprehensive Cancer Network (NCCN) in the USA recommend SLNB, provided that dual tracers are used and a minimum of three sentinel nodes are removed [9,10]. In countries such as Italy, Denmark, Russia, and Hungary, SLNB or TAD are accepted as first choice for axillary staging in this group of patients. The German Breast Committee of the Working Group for Gynaecological Oncology (AGO Breast Committee) endorses both TAD and ALND as recommended strategies [11].

Unanswered questions include the role of axillary imaging for selection of patients who might safely be offered surgical de-escalation [12]. Further, the necessity of regional therapy, e.g., radiotherapy following axillary staging in patients with pathological complete response on SLNB or TAD, is still a matter of debate.

In order to shed light on this much debated topic, the European Breast Cancer Research Association of Surgical Trialists (EUBREAST) has initiated AXSANA (AXillary Surgery After NeoAdjuvant Treatment), a multinational prospective cohort study (NCT04373655) which enrolls cN+ patients undergoing NACT who convert to ycN0. The aim of AXSANA is to assess the impact of different surgical staging procedures in the axilla on the oncologic outcome and on health-related quality of life.

## 2. Targeted Axillary Dissection: More Questions Open Than Answered

### 2.1. Which Marking Technique Is Optimal?

So far, several methods for marking of target lymph node(s) have been developed, usually based on techniques that are already in use for the localization of non-palpable breast lesions (Table 4). Interestingly, there are notable regional differences regarding the use of various techniques. The same method may be the technique of choice in one country, while being completely unknown in another.

To date, the largest amount of data has been published on clip-based targeted axillary dissection (TAD). Unless intraoperative ultrasound is used, this strategy requires a preoperative localization step, performed either by the use of a wire or by placing another marker (e.g., magnetic or radioactive seed, radar marker or radiofrequency identification [RFID] tag) into the clipped area that will allow identification during surgery. Still, the success of target lymph node (TLN) removal depends on the ultrasound visibility of the clip inserted before NACT.

The study that brought international attention to the technique was a retrospective analysis of a prospective database at the M.D. Anderson Cancer Center (Table 5) [8]. Nearly all patients in this study underwent ultrasound-guided placement of a radioactive iodine-125 seed into the previously clipped node prior to surgery. This strategy offers more flexibility than wire localization, which was used in two patients only, because the seed can be inserted several days before surgery, whereas the wire placement is usually scheduled for the morning before the operation or (rarely) the day before. The study by Caudle et al. showed that the FNR of TAD can be as low as 2.0% [8]. However, since only patients with successful preoperative localization were included in this analysis, it is unclear whether the clip could not be visualized in some patients, making preoperative localization impossible.

The results of the prospective German multicenter SENTA register study were presented at the ESMO conference in 2019 and are now available as a full publication [28,29]. In this study, the TLN was clipped before NACT in 473 patients. A Tumark Vision clip (SOMATEX®, Berlin, Germany) was used in the majority of cases (71%), followed by an O-Twist clip (BIP, Türkenfeld, Germany) (12%). Clip types used in the remaining patients have not been described. In 50 out of 473 patients, a targeted lymph node biopsy (TLNB) was not attempted either because the clip was not visible upon preoperative ultrasound or TLNB was for some reason not planned by the investigator. The TLN could be successfully removed in 329 of 423 patients, mostly after wire localization, resulting in an overall removal rate of 78% among patients in whom a TLNB/TAD was attempted. The TLN detection rates were higher in case a Tumark Vision or a O-Twist clip was used, compared to other clips (79.7%, 78.4%, 69.4%, respectively). Interestingly, triple negativity and high Ki67 index were associated with TLNB failure (i.e., TLN not removed during TLNB attempt). In 63% of patients, the SLN and the TLN were identical. In the subgroup of 278 patients who received ALND, TLNB alone and TAD resulted in FNRs of 7.2% and 4.3%, respectively.

Another marking method is the injection of a small volume (0.1–0.5 mL) of a carbon solution into the suspicious lymph node. At surgery, the TLN is identified visually by its dark staining. The results of TATTOO, the largest prospective trial on carbon solution-based TAD, have been presented recently [47]. In 94% of 118 included patients, the marked node could be detected intraoperatively. In 60% of cases, SLN and TLN were identical. In 5 (4.5%) patients, unintentional tattooing of the skin was observed, but it remains unclear whether this effect is permanent. In 61 cN+ patients converting to ycN0, completion ALND was performed. In this subgroup, three out of 33 patients with residual axillary disease had not been correctly identified by TAD, resulting in a FNR of 9.1%. A similar FNR of 8.3% was reported from the only other study on FNR in carbon solution-based TAD [50].

The MARI technique (“Marking the Axillary lymph node with Radioactive Iodine seeds”) was reported from the Netherlands in 2015 [44]. It implies that a titanium-encapsuled radioactive seed is inserted into the suspicious lymph node before NACT under ultrasound guidance. In contrast to technetium-99-based localization of non-palpable breast lesions before primary surgery, the longer half-life (59 days) of iodine-125 allows for use in the neoadjuvant setting. Donker et al. evaluated the MARI technique in 100 patients scheduled for NACT [44]. Marking of the biopsy-proven positive lymph node and the breast tumor using identical seeds was conducted at one procedure. In three patients, the seed could not be properly positioned in the lymph node, resulting in a removal rate of the biopsied node of 97%. No relevant loss of signal was observed after a median NACT duration of 17 weeks, and all seeds could be removed successfully. Importantly, no SLNB was performed in this study, and thus the reported FNR of 7% refers to the TLNB alone.

Two aspects have been critically discussed after the introduction of the MARI technique. First, the procedure requires complex safety regulations and cannot be performed in some countries due to national radiation protection laws. Secondly, both the localization of iodine-125 and technetium-99 used for SLN mapping require a gamma probe so interference might potentially occur in case of down-scatter of technetium-99m into the energy spectrum of iodine-125. The first results of the RISAS trial (NCT02800317), presented by Simons et al. at the San Antonio Breast Cancer Symposium in 2020 [53], could, however, show that the combination of the MARI technique and SLNB is feasible: TAD was successfully performed in 223 out of 227 (98%) patients. The trial was designed to test the non-inferiority of TAD over ALND, with FNR as the main endpoint. Non-inferiority was assumed if the upper bound of a 95% confidence interval around the FNR was lower than 6.25%. Although the FNR was low (3.5%), the prespecified primary endpoint was not met (95% confidence interval 1.38–7.16). The authors reported, however, that 4 out of 5 patients with a false-negative result (i.e., residual lymph node metastasis in the ALND specimen despite tumor-free TAD) were found among the first 10 enrolled patients of participating institutions, suggesting a learning curve that should be considered when planning future studies.

Another technique to mark a biopsy-proven lymph node metastasis is based on the placement of a magnetic seed. So far, the only magnetic seed for axillary node marking evaluated in studies is the MagSeed^®^ (Endomag, London, United Kingdom). The seed is detected during surgery by using a magnetic probe called Sentimag^®^ (Endomag, London, United Kingdom) surgical guidance platform. This method has already provided evidence for the successful resection of non-palpable breast lesions [54]. Its use for TAD was restricted by the initial requirement to remove seeds within 30 days after placement, but recently, both the Food and Drug Administration (FDA, 2018) and the Conformité Européenne (CE) marking (2020) approved long-term implantation in any soft tissue, thus allowing use in the neoadjuvant setting. First data on the magnetic seed-based TAD have been recently published by Thill et al. [45]. This small study showed a detection rate of 100%. Since magnetic seeds can lead to large magnetic resonance imaging (MRI) artifacts, it should be used with caution in patients in whom MRI-based assessment of response during NACT is planned.

Other possible techniques are radar reflector-localization clips [55] and radiofrequency identification (RFID) tags [56,57], that have provided promising results for non-palpable lesion localization, while data on lymph node marking are limited. Table 4 shows potential advantages and disadvantages of various methods and Table 5 gives an overview of current evidence.

### 2.2. How Many Nodes Should Be Marked?

Since TAD is a relatively new technique that still needs to be validated and standardized, various institutions follow different strategies. So far, there is no consensus on the number of nodes that should be marked in patients with more than one suspicious lymph node on imaging. While most studies report the marking of a single node, it is unclear which lymph node was selected for biopsy and marking in these cases. Options vary between the largest, the “most suspicious”, or the best accessible lymph node [8,47]. Other authors report marking of all suspicious nodes [42,48]. Both strategies offer their own advantages and disadvantages (Table 6).

Regarding the accuracy of lymph node assessment, marking only one axillary lymph node in case of patients with several suspicious or biopsy-proven nodes might increase the FNR since the marked lymph node might be free of tumor while other previously suspicious but not marked lymph nodes may still contain tumor residuals and remain in situ. A heterogenous axillary response to NACT is common and can be diagnosed in up to 74% of patients with hormone receptor-positive HER2-negative disease, followed by 29% in triple-negative and 25% in HER2-positive tumors [58]. Lim et al. conducted a small study with a complex design to explore FNRs of different TLNB strategies in the same set of patients [42]. All suspicious nodes of a patient were marked with different types of clips to aid individual node identification (e.g., UltraCor Twirl, HydroMARK, and UltraClip). Altogether, 21 nodes in 14 patients were clipped. After NACT, all patients received TLNB and ALND. The first clipped node alone had a FNR of 7.1%, which sank to 0% when the second clipped node was taken into account. No SLNB was performed, and it remains unclear how this might have influenced the FNR.

Concerning arm morbidity, marking all suspicious nodes will necessarily result in removal of more lymph nodes. Natsiopoulos et al. conducted carbon solution-based TAD (Spot^®^, GI Supply, Inc., Mechanicsburg, PA, USA) in 75 patients [48], marking each biopsy-proven or strongly suspicious node (median 2, range 1–5). At surgery, 2–10 nodes (median 4) were retrieved. How this relatively high number of removed nodes may affect quality of life and arm morbidity remains to be clarified. Importantly, a clinically relevant balance between high accuracy (more extensive staging) and low morbidity (less extensive staging) must be found.

### 2.3. When Should Lymph Nodes Be Marked?

To date, there is no consensus on the optimal timepoint of marker placement. In most studies, lymph node metastases were confirmed by fine needle aspiration or core biopsy. Such procedures, however, are not standard in some countries, and not all guidelines recommend routine minimally invasive biopsy in case of suspicious findings. Especially in case of large tumors and multiple highly suspicious nodes, clinicians may find it sufficient to perform biopsy of the breast tumor only.

If a minimally invasive biopsy is performed, the marker may be inserted into the same lymph node(s) at the same session or at a second session upon confirmation of metastasis by pathology/cytology. Data on this aspect lack detail in available studies but most authors report the placement of the marker into the “previously proven” node [44,48]. In contrast, marking of the suspicious lymph node immediately after the biopsy is reported by others [31,52]. Table 7 provides an overview of advantages and disadvantages of both strategies.

### 2.4. What to Do in Case of a “Lost Marker”?

A major concern expressed by clinicians trying to implement TAD at their institutions is the uncertainty of how to deal with patients in whom the marker has not been retrieved at surgery. Since a TAD/TLNB cannot be successfully performed in these patients, completion ALND is an obvious choice. Still, in some patients, the marker will not be found in the ALND specimen either. The pivotal study by Caudle et al. included 208 patients whose metastatic nodes were clipped prior to NACT [8]. In five of these patients, the clip could neither be identified in the surgical specimen nor upon radiography of the axilla, suggesting clip dislodgement. In another study, the clip could not be retrieved in two out of 73 patients, but no further details were reported [30]. To date, the management of patients with lost clips is unclear.

In case of other markers, several additional concerns need to be addressed. Since leaving radioactive seeds behind results in a major radiation regulation breech that must be reported to regulation authorities, seed explantation should be attempted whenever possible unless it would jeopardize patient’s well-being [59]. In case of magnetic seeds, no radioactivity is involved, but leaving a magnetic marker in the axilla can result in large MRI artifacts and thus compromise imaging assessment during post-treatment surveillance. Minor MRI artifacts are also possible in case of unsuccessful radar marker or RFID tag retrieval.

While lost clips can only be removed using imaging-guidance (usually radiography, ultrasound, or computer tomography), radioactive and magnetic seeds as well as radar markers and RFID tags can be identified using a special probe and carbon ink-marked nodes can only be visualized intraoperatively during dissection of the axilla, possibly resulting in different radicality of retrieval procedure and individual risk faced by the patient.

### 2.5. Is TAD Safe for All Patients?

It is yet unclear whether all cN+ patients can safely omit completion ALND in case of negative axillary staging. So far, while none of the studies on TAD have reported oncological outcome or health-related quality of life, there is limited information available on SLNB alone in this setting. Kahler-Ribeiro-Fontana et al. have recently reported on long-term outcomes following SLNB after NACT [3]. Among 222 clinically node-positive patients, 123 had no residual axillary disease at surgery. Of these, only two patients (1.6%) developed axillary recurrence after 3.6 and 5.5 years from surgery and were alive without disease at the last follow-up. Among all cN1/2 patients, no significant differences in 5-year and 10-year overall survival were found between patients who received ALND and those in whom ALND was omitted.

Most TAD validation studies include heterogeneous groups of patients. In the pivotal study by Caudle et al., nearly all patients had cT1–cT3 tumors, but 28% of patients presented with at least four abnormal nodes on ultrasound [8]. Similarly, in the largest study on carbon solution-based TAD, 25% of patients had ≥4 and 4% had ≥10 suspicious nodes before NACT [47]. 124 out of 423 (29%) patients enrolled in the SENTA trial had at least three abnormal nodes [29]. Intuitively, the more nodes appear suspicious on initial ultrasound, the more probable it is for the TAD to miss residual disease, especially in case only one node was marked, thus resulting in a higher FNR [60].

Other factors that can improve identification of patients who do not benefit from completion ALND are breast response to therapy and tumor subtype. A multivariable analysis of 13,396 patients showed that pathological complete response (pCR) n the breast was the most important predictor of pCR in the axilla (odds ratio 20.37 for yT0) [61]. Other studies reported a strong association between response in the breast and in the axilla, particularly in patients with HER2-positive and triple-negative breast cancer [62,63,64], implying that tumor subtype should probably also be implemented into a potential decision algorithm. Since patients with HER2-positive and triple-negative cancer achieving radiological breast response have the highest probability of reaching axillary pCR, they are probably most likely to benefit from de-escalation of surgical treatment, such as TAD.

### 2.6. Beyond Surgical Therapy: Which Fields Should Be Irradiated after TAD?

Another controversial issue is the target volume for nodal irradiation. While some guidelines exclude levels I and II from irradiation in case of a negative TAD, others suggest to cover all regions if they have not been assessed with a therapeutic intent. More specifically, it remains unclear whether patients with an extensive axillary tumor burden who achieve pCR after NACT benefit from additional regional treatment.

Koolen et al. from the Netherlands Cancer Institute proposed a hypothetical algorithm based on initial nodal status and response to NACT [65]. In this algorithm, patients receive not only ultrasound but also PET-CT prior to NACT. Those with 1–3 positive nodes are recommended no further axillary therapy in case of axillary pCR, defined as a negative MARI procedure (i.e., resection of a radioactive seed-marked TLN) [44]. In case of non-pCR, axillary radiotherapy is performed. For patients with ≥4 positive nodes before NACT, axillary radiotherapy is always indicated but patients with non-pCR are also recommended an ALND. In a cohort of 100 patients treated by TLNB (MARI) and ALND, the proposed algorithm would have led to omission of ALND in 74%, and some patients potentially risk under- (3%) or over-treatment (17%) [65]. Whether such tailored strategies might be an acceptable compromise between high oncological safety on one side and lower arm morbidity on the other remains to be clarified in future trials. In this context, the results of the NSABP-B51 trial (NCT01872975) are expected to be published in 2023. This phase III clinical trial is designed to test whether regional nodal irradiation (RNI) improves the recurrence-free interval rate in women with cN1 breast cancer before NACT who become pathologically node-negative at the time of surgery. Patients who undergo mastectomy are randomly assigned to observation or radiotherapy to the chest wall and undissected axilla, internal mammary nodes, and ipsilateral supraclavicular fossa, whereas women who undergo breast-conserving surgery are randomized to adjuvant whole breast irradiation vs. whole breast and regional nodal irradiation. In the ALLIANCE A011202 (NCT01901094) trial, women with node-positive status before NACT receive SLNB at the time of surgery. Patients with axillary residual disease are randomized to completion ALND and RNI vs. RNI alone.

## 3. The AXSANA Study: Which Axillary Strategy Is Optimal in the cN+ → ycN0 Setting?

AXSANA, initiated by EUBREAST (http://axsana.eubreast.com; accessed on 27 March 2021, Figure 1), is a large, prospective, non-interventional cohort study aiming to evaluate the role of axillary treatment in cN+ patients undergoing NACT. With a target accrual of 3000 patients, the study is expected to be able to resolve several open issues. Patients with clinically positive nodal status at time of diagnosis who are scheduled to receive axillary surgery after NACT can be enrolled. Axillary staging procedures and treatment modalities are chosen at the discretion of the treating physicians and according to national and institutional guidelines. Inclusion and exclusion criteria are presented in Table 8. Follow-up is annually during the first five years after surgery. Arm morbidity and quality of life are evaluated at baseline and after 1, 3, and 5 years, using four validated questionnaires (EORTC QLQ-C 30, EORTC QLQ BR 23, Lymph-ICF, and Sense of Coherence). Financial support has been provided by the AGO-B study group, the AWOgyn (German Working Group for Reconstructive Surgery in Oncology-Gynecology), Claudia von Schilling Foundation for Breast Cancer Research, EndoMag, Mammotome, and MeritMedical. AXSANA is further supported by the North-Eastern German Society of Gynecological Oncology (NOGGO) and the German Breast Group (GBG).

Primary study endpoints:5-year invasive disease-free survival (iDFS)3-year axillary recurrence rateQuality of life and arm morbidity

Secondary study endpoints:Feasibility of different axillary staging strategies assessed by detection rates for SLN and/or TLNSuccess rate of nodal staging using different axillary staging techniquesNumber of removed lymph nodes and their association to complications, arm morbidity, and quality of lifeOperating time as a surrogate parameter for surgical resourcesProportion of node-positive patients according to the strategy used (as a surrogate parameter for FNR)Factors associated with successful detection of the TLNImpact of learning curve on success rates of TADSurgical standards of care in different European countriesTreatment decisions in case of ypN+ status following NACT (ALND vs. radiation therapy)iDFS in patients with ypN+ status who received ALND or radiotherapy or bothAnalysis of factors contributing to a decreased quality of life and subjective symptoms of arm morbidity, i.e., baseline quality of life and sense of coherence, extent of axillary surgery, and other locoregional and systemic therapies receivedEconomic resources required for different axillary staging strategies and techniques (material costs, operating time, etc.)

The first AXSANA study site was opened in Germany in June 2020, and recruitment began the same month. Currently, there are 328 patients enrolled. Twenty countries are participating in the study, most of which are in the process of applying for ethical approval. Ten countries have at least one study site open (Figure 2). The study will hopefully address several unanswered issues, such as:Which staging technique should be recommended to cN+ patients converting to ycN0?Is imaging helpful in identifying patients most likely to achieve pCR in the axilla? If yes, which method should be recommended?Should cN1 and cN2/3 patients be offered different surgical strategies for axillary staging?Which lymph node marking technique offers highest rates of successful TAD/TLNB?

Due to high complexity and discordant recommendations, a randomized trial comparing different techniques seems hardly feasible and therefore would not clarify currently open issues within a reasonable timeframe. In the AXSANA study, patients are treated at physicians’ discretion. To allow comparisons between different cohorts (SLNB, TLNB, TAD, ALND), detailed data regarding clinical and pathological parameters are obtained.

Other currently ongoing studies investigating axillary management in the neoadjuvant setting in cN+ patients are presented in Table 9.

## 4. Conclusions

In view of continuously improving primary systemic treatments with increasing response rates, there is an urgent need to adapt and de-escalate strategies for axillary surgery, since it is strongly associated with postoperative morbidity. While SLNB has successfully replaced ALND as a staging procedure in primary surgery and after NACT for patients with an initial cN0 status, there is an ongoing debate on appropriate axillary staging for patients who convert from cN+ to ycN0.

This review revealed heterogeneous guideline recommendations and practice throughout the international community. This observation is explained by the lack of evidence concerning minimally invasive staging procedures like SLNB or TAD and their association with oncological outcomes, arm morbidity, and quality of life. The review also identified a multitude of unresolved issues regarding indications, surgical staging procedures, and technical aspects of lymph node marking. As a consequence, the European Breast Cancer Research Association of Surgical Trialists (EUBREAST) initiated a prospective cohort study that will allow the assessment of clinically relevant parameters in different axillary staging procedures after NACT, and of many open issues that have been highlighted here. AXSANA is open for all countries provided that patients receive treatment according to current international standards (http://axsana.eubreast.com, accessed on 27 March 2021).

## Figures and Tables

**Figure 1 cancers-13-01565-f001:**
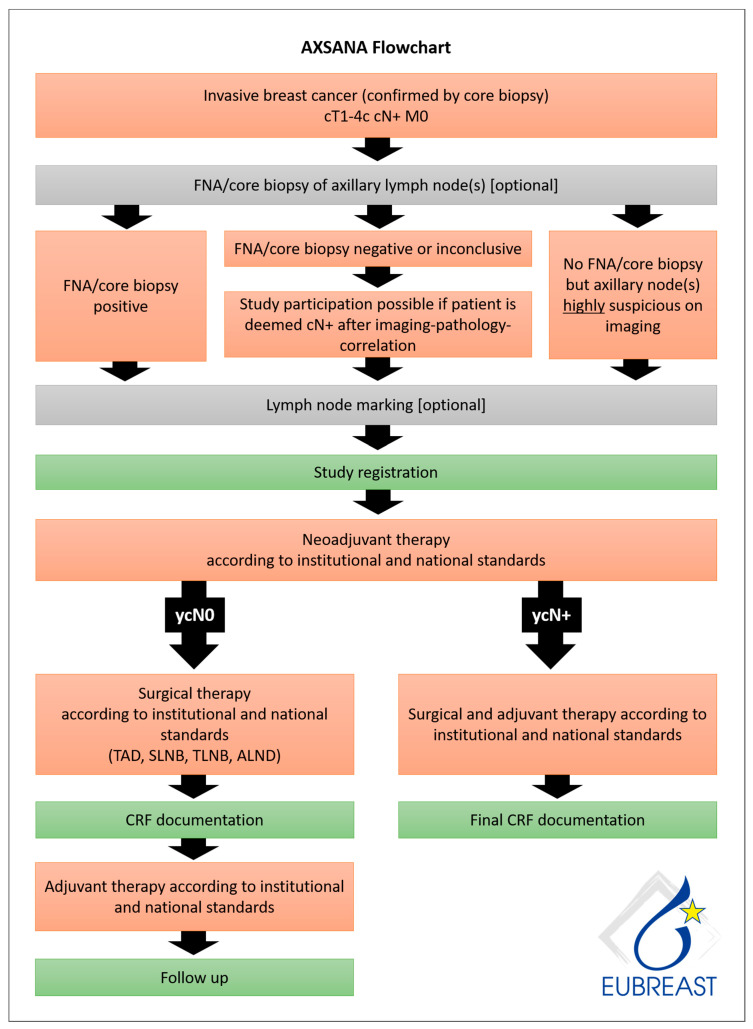
AXSANA flow chart.

**Figure 2 cancers-13-01565-f002:**
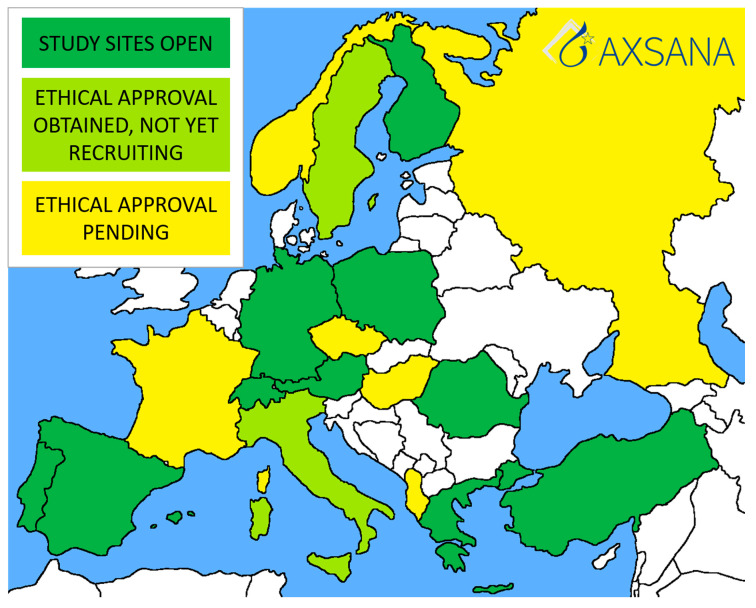
Current status of the AXSANA study.

**Table 1 cancers-13-01565-t001:** Axillary surgical staging techniques: The most important definitions.

Type of Surgery	Description
Axillary lymph node dissection (ALND)	Systematic removal of lymph nodes from the axilla, usually level I and II, sometimes including also level III
Sentinel lymph node biopsy (SLNB)	Identification and removal of the sentinel lymph node, usually using radioactive tracer (Technetium-99) or blue dye
Targeted lymph node biopsy (TLNB)	Selective removal of metastatic lymph node(s) marked before neoadjuvant therapy
Targeted axillary dissection (TAD)	Combination of TLNB and SLNB

**Table 2 cancers-13-01565-t002:** National and international guidelines on axillary surgical staging in initially node-positive patients receiving neoadjuvant therapy.

National/International:	Staging Recommendation for cN+ → ycN0 Patients	Level of Evidence/Grade of Recommendation
European Society for Medical Oncology (ESMO) [10]	Sentinel lymph node biopsy (SLNB) can be an option, as long as additional recommendations are followed (e.g., dual tracer, clipping/marking of positive nodes, minimum of three sentinel nodes removed)	III, B
National Comprehensive Cancer Network (NCCN) [9]	Consider SLNB. Relatively high false-negative rate (FNR) (>10%) can be improved by marking biopsied lymph nodes to document their removal, using dual tracer, and by removing more than 2 sentinel nodes	2B
American Society of Breast Surgeons [13]	If SLNB after neoadjuvant therapy is attempted, dual tracer should be used. If a SLN and/or the clipped node (if clipped) is not identified, an Axillary lymph node dissection (ALND) is recommended	Not provided
Finland [14]	ALND	Not provided
Germany (S3 guideline) [15]	ALND	2b, B
Germany (AGO Breast Committee) [11]	Targeted axillary dissection (TAD): + (i.e., this investigation or therapeutic intervention is of limited benefit for patients and can be performed) ALND: + (i.e., this investigation or therapeutic intervention is of limited benefit for patients and can be performed) SLNB only: +/− (i.e., this investigation or therapeutic intervention has not shown benefit for patients and may be performed only in individual cases. According to current knowledge a general recommendation cannot be given)	2b, B
Hungary [16]	SLNB, preferably with double tracer technique (isotope + dye), and with at least 3 SLNs removed; in case of limited axillary tumor load and a realistic chance of cN1 → ycN0 conversion, clipping the metastatic node before neoadjuvant chemotherapy (NACT) is recommended	Not provided
India [17]	No specific recommendation for cN+ ycN0 patients	Not provided
Poland [18]	SLNB can be an option with some limitations: Remove ≥ 3 SLN if nodes were not clipped/marked; if not fulfilled, → ALND [2+] Dual tracer (radiocolloid and patent blue) [2+] Additional option is clipping/marking lymph nodes before NACT [0] Remove all clipped lymph nodes and SLNs, if not fulfilled → ALND [2+] For identification of clipped nodes intraoperative ultrasound or guidewire is recommended [0] Techniques with ferromagnetic tracer [0]	Power of recommendation in square brackets (score −2, −1, 0, 1+, 2+)
Romania	Last approved national guideline (2009) [19]: ALND is recommended, SLNB is not recommended after NACT New version proposed by the Romanian Society of Obstetrics and Gynecology (2019; not approved by the Ministry of Health) [20]: Suspicious lymph nodes must be biopsied, and clipped if possible; if SLNB after NACT is attempted, dual tracer is recommended	Not provided
Sweden [21]	ALND	Grade +++ Recommendation: B
**Society Guidelines**		
Denmark (Danish Breast Cancer Cooperative Group) [22]	TAD including double tracer technique (radioactive tracer plus dye) Target lymph node(s) to be marked with radioactive iodine seeds or coils	Not provided
Italy (Assoziacione Italiana de Oncologica Medica = AIOM) [23]	SLNB; ALND omission may be considered in the case one or more negative sentinel lymph nodes, identified with double tracer and only in patients who were cN1/2 at time of diagnosis	Quality of evidence: Low Strength of Recommendation: Weak
Portugal (Portuguese Society of Senology) [24]	cN1 patients should be clipped and ycN0 patients should be managed by TAD, with omission of ALND in ypN0 if the following criteria are fulfilled: (1) SLNB performed using dual traced, (2) clipped node removed, and (3) more than 2 removed nodes	Not provided
Russia (Association of Oncologists of Russia) [25]	It is recommended to mark the tumor before starting neoadjuvant therapy to enable visualization during subsequent surgical treatment. If it is impossible to perform SLNB or if a metastatic focus in the SLN is detected, it is recommended to perform ALND.	III, B I, A
Spain (Spanish Society of Medical Oncology) [26]	ALND is recommended. In selected cN+ cases, in which positive axillary node has been marked prior to NACT, the identification and recovery of >2 negative SLNs (including the marked node) with a double tracer technique (Tc99 and methylene blue) may avoid ALND.	I, A II, C

**Table 3 cancers-13-01565-t003:** Studies on sentinel node biopsy after neoadjuvant therapy in initially node-positive patients.

Study	Number of Patients	Preoperative Axillary Assessment	Detection Rate of the Sentinel Node	False Negative Rate
SENTINA [4]	592	Clinical examination, ultrasound	80.1%	14.2%
SN FNAC [27]	153	Clinical examination, ultrasound	87.6%	8.4% ^1^
ACOSOG Z1071 [5]	649	Surgical approach independent of clinical response	92.9%	12.6% ^2^
GANEA 2 [1]	307	Surgical approach independent of clinical response	79.8%	11.9%
Meta-analysis [6]	3398	-	91%	13%

^1^ Sentinel nodes with isolated tumor cells [ypN0(i+)] defined as positive. ^2^ Only in patients with at least 2 sentinel nodes removed (pre-defined study criterion); in case of only one sentinel node removed, the false negative rate was 29.3% [7].

**Table 4 cancers-13-01565-t004:** Possible options for marking and localizing suspicious lymph nodes prior to start of neoadjuvant chemotherapy (modified after Reference [12]).

Marking	Localization	Advantages	Disadvantages
**Clip**	Preoperative imaging-guided wire localization (mostly ultrasound-guided) Intraoperative ultrasound Preoperative placement of a radioactive/magnetic seed, radar marker, or ink into the clipped area (mostly ultrasound-guided)	Largest amount of data Reliable radiographic visibility No radioactivity involved Relatively low cost	Visibility on ultrasound varies widely between studies, and a large part of the axilla is not visible on a mammogram Preoperative localization necessary (wire/seed) unless intraoperative ultrasound is used Results from studies comparing different clips not yet available Relatively low detection rate (rate of successful target lymph node (TLN) removal 70% in the largest available dataset [28]) Visibility of some clips (e.g., hydrogel clips) may decrease over time Reaction of the node tissue to the clip (especially hydrogel-containing clips) may be misinterpreted on pathological examination Some clips approved explicitly for marking in the breast, not in the axilla Allergic reactions rare but possible (some titanium clips contain nickel)
**Radioactive seed**	Intraoperative localization using gamma probe	High detection rate No preoperative wire localization necessary Transcutaneous localization before skin incision possible	Procedure not authorized in some countries, requires complex radiation safety procedures Signal reduction over time (i.e., in case of longer chemotherapy due to interruptions) High cost Allergic reactions rare but possible (some seeds contain nickel)
**Carbon suspension**	Intraoperative visualization	No preoperative wire localization necessary No radioactivity involved Low cost	Limited data Marking cannot be localized without surgical exploration of the axilla Possible ink migration Possible skin discoloration In case blue dye is used for SLNB, the ink colors must differ
**Magnetic seed**	Intraoperative localization using magnetic probe	No preoperative wire localization necessary No radioactivity involved Transcutaneous localization before skin incision possible	Very limited data Concerns regarding use in patients with pacemakers and implantable defibrillators Standard metal surgical tools should not be used during measurement Allergic reactions rare but possible despite very low nickel content MRI artifacts High cost Localization in deep tissue may result in weaker signal (recommended depth max. 3.5 cm)
**Radar reflector localization (RRL)**	Intraoperative localization using radar locator	No preoperative wire localization necessary No radioactivity involved Transcutaneous localization before skin incision possible	Very limited data High cost Allergic reactions rare but possible (some markers contain nickel) Minimal MRI artifacts possible Interference with older halogen lights in the operating theatre possible Adequate localization may be limited in case of a large distance between marker and detection probe
**Radiofrequency identification devices (RFID tags)**	Intraoperative localization using radiofrequency localizer	No preoperative wire localization necessary No radioactivity involved No decrease of signal over time Transcutaneous localization before skin incision possible	Very limited data High cost MRI artifacts possible Concerns regarding use in patients with pacemakers and implantable defibrillators

**Table 5 cancers-13-01565-t005:** Marking and localization methods for target lymph node retrieval in breast cancer patients undergoing neoadjuvant chemotherapy.

Marking Technique before NACT	Trial	Number of Patients	Localization Technique	Preoperative or Intraoperative Detection Rate of the Marker	Successful TLN Removal	FNR ^1^
**Clip**	SENTA [28,29]	473	Preoperative wire localization in most patients	Ultrasound: 89%	78%	TAD: 4.3% TLNB: 7.2%
	Caudle 2016 [8]	208	Preoperative radioactive seed placement into the clipped area → intraoperative detection using gamma probe	NR	98% ^2^	Clipped node removal: 4.2% ^3^ Clipped node removal + SLNB: 2.0% SLNB alone: 10.1%
	ACOSOG Z1071 [5,7]	203	None	NR	83% ^4^	SLNB: 6.8% if TLN was SLN 19% if TLN was not SLN ^5^
	Plecha 2015 [30] (in 98% HydroMARK clip)	91	Wire localization in 74% of patients	NR	97% in patients with and 83% in patients without wire localization	NR
	Laws 2020 [31]	57	Preoperative placement of a magnetic seed, a RRL clip or a RFID tag into the clipped area	98%	89%	NR
	Ngyuen 2017 [32]	56	Preoperative radioactive seed placement → intraoperative detection using gamma probe	Ultrasound: 72%	91%	NR
	Simons 2021 [33]	50	Preoperative magnetic seed placement → intraoperative detection using magnetic probe	Ultrasound: 100%	98%	NR
	ILINA trial [34] (HydroMARK clip)	46	Intraoperative ultrasound	Ultrasound: 96%	NR	TAD: 4.1% ^3^
	Sun 2020 [35]	38	Preoperative RRL clip placement → intraoperative detection using radar probe	100%	100%	NR
	Hartmann 2018 [36] (HydroMARK clip)	30	Wire localization in 80% of patients (67% US, 13% mammography)	Ultrasound: 83%	70% in the entire cohort, 83% in patients with wire localization	0%
	Diego 2016 [37]	30	Preoperative radioactive seed placement into the clipped area → intraoperative detection using gamma probe	Ultrasound: 93%	93%	NR
	Mariscal Martinez 2021 [38] (HydroMARK 93%, Tumark 3%, UltraCor-Twirl 3%)	29	Preoperative magnetic seed placement → intraoperative detection using magnetic probe	100%	100%	SLNB alone: 21.4% TAD: 5.9%
	Kim 2019 [39] (UltraClip)	28	US-guided injection of ink and skin marking	Ultrasound: 79% clearly visible, 21% equivocally visible	96%	NR
	Balasubramanian 2020 [40] (HydroMARK clip)	25	Wire localization	NR	92%	NR
	Lim 2020 [41,42]	14	Preoperative US-guided skin marking	NR	84% (UltraCor Twirl: 100% HydroMARK: 78% UltraClip DualTrigger: 50% UltraClip: 0%)	TLNB: 0% if ≥2 marked nodes were removed 7.1% if only first TLN was removed
**Radioactive seed**	RISAS [43]	227	Gamma probe (intraoperative)	NR	98%	TAD: 3.5%
	Donker 2015 [44]	100	Gamma probe (intraoperative)	100% (gamma probe)	97%	TLNB: 7% ^3^
**Magnetic seed**	Thill 2020 [45]	5	Magnetic probe (intraoperative)	100%	100%	NR
**Radar reflector localization-clip**	Sun 2020 [35]	7	Intraoperative radar localization	100%	100%	NR
**RFID tag**	Malter 2020 [46]	10	Radiofrequency probe (intraoperative)	100%	100%	NR
**Carbon suspension**	Hartmann 2020 [47]	118	Intraoperative visualization	94%	94%	TAD: 9.1%
	Natsiopoulos 2019 [48]	75	Intraoperative visualization	100%	100%	NR
	Allweis [49]	63	Intraoperative visualization	95%	95%	NR
	Khallaf 2020 [50]	20	Intraoperative visualization	95%	95%	TAD: 8.3% SLNB alone: 15.3%
	Gatek 2020 [51]	20	Intraoperative visualization	100%	100%	NR
	Choy 2014 [52]	12	Intraoperative visualization	100%	100%	NR

^1^ Analyzed only in patients receiving ALND. ^2^ The clip was absent on postoperative axillary radiography in the remaining five patients, suggesting clip dislodgement. ^3^ Lymph nodes with isolated tumor cells were considered positive. ^4^ In the remaining 17% of patients, the clip was neither in the SLN nor in the ALND specimen. ^5^ Only in patients with ≥2 SLNs removed and initially cN1. Abbreviations: ALND—axillary lymph node dissection, FNR—false-negative rate, MARI—marking the axillary lymph node with radioactive iodine (^125^I) seeds, NACT—neoadjuvant chemotherapy, NR—not reported, RFID—radiofrequency identification device, RRL—radar reflector localization, SLNB—sentinel lymph node biopsy, SLN—sentinel lymph node, TAD—targeted axillary dissection (removal of marked node and SLN), TLN—target lymph node, TLNB—target lymph node biopsy, US—ultrasound.

**Table 6 cancers-13-01565-t006:** Potential strategies regarding the number of marked nodes.

	Only One Node Is Marked	All Suspicious Nodes Are Marked
**Advantages**	Lower cost Fewer nodes are removed at surgery → possibly less arm morbidity Less challenging marking procedure	Lower FNR in small studies → possibly better oncological outcome
**Disadvantages**	Heterogenous response of different nodes to therapy → higher FNR → possibly higher recurrence rate	High cost Higher probability that one of the marked nodes will not be removed successfully More nodes need to be removed → arm morbidity Complicated marking procedure

**Table 7 cancers-13-01565-t007:** Potential strategies regarding the time point of lymph node marking.

	At Time of Biopsy	After Pathological/Cytological Confirmation of Nodal Metastasis
**Advantages**	Only one invasive procedure for the patient Certainty that the marking has been placed into the biopsied node	The marker is placed only if necessary, i.e., in case a TLNB/TAD is planned
**Disadvantages**	Some lymph nodes might be marked unnecessarily → higher cost	In case of several suspicious nodes, the marker may be placed into the one that has not been biopsied In case of reactive lymph nodes due to biopsy, the marker may be placed into a benign node An additional invasive procedure is necessary

**Table 8 cancers-13-01565-t008:** The AXSANA study: Inclusion and exclusion criteria.

Inclusion Criteria	Exclusion Criteria
Signed informed consent form Primary invasive breast cancer (confirmed by core biopsy) cN+ (confirmed by core biopsy/fine needle aspiration or presence of highly suspicious axillary node(s) on imaging) In case a minimally invasive biopsy of axillary lymph node(s) has been performed and yielded a negative or inconclusive result, patients may be included if the final classification after imaging-pathology-correlation is cN+ cT1–cT4c Scheduled for neoadjuvant systemic therapy Female/male patients ≥ 18 years old	Distant metastasis Recurrent breast cancer Inflammatory breast cancer Extramammary breast cancer Bilateral breast cancer History of invasive breast cancer, ductal carcinoma in situ, or any other invasive cancer Confirmed or suspected supraclavicular lymph node metastasis Confirmed or suspected parasternal lymph node metastasis Axillary surgery before NACT (e.g., SLNB or nodal sampling) Pregnancy Less than 4 cycles of NACT administered Patients not suitable for surgical treatment

**Table 9 cancers-13-01565-t009:** Current trials investigating de-escalation of surgical treatment in cN+ patients undergoing neoadjuvant therapy.

Study	Status	Study Design	Primary Endpoint(s)
**ATNEC** NCT04109079	Not yet recruiting	Randomized trial cN+ → ypN0 patients Surgery: TAD Marking technique: clip or carbon suspension Arms: axillary treatment (ALND or ART) vs. no axillary treatment	DFS Patient-reported lymphedema
**AXSANA** NCT04373655	Recruiting since June 2020	Non-interventional cohort study cN+ patients	iDFS Axillary recurrence rate Quality of life and arm morbidity
**GANEA 3** NCT03630913	Recruiting since January 2019	Single-arm trial cN+ patients (confirmed by biopsy) Surgery: TAD followed by ALND Marking technique: clip, marking of the most suspicious node only	False-negative rate of SLNB, TLNB, and TAD
**MAGELLAN** NCT03796559	Recruiting	Single-arm trial cN+ patients (confirmed by biopsy) Surgery: TAD Marking technique: clip and magnetic seed	Retrieval rate of clipped node and magnetic seed
**Pre-ATNEC** NCT03640819	Completed, results pending	Single-arm trial cN+ patients (confirmed by biopsy) Surgery: TLNB and—at surgeon’s discretion—SLNB or ALND Marking technique: carbon suspension	Identification rate of marked lymph node(s)
**RISAS** NCT02800317	Completed, full publication pending [43,53]	Single-arm trial cN+ patients (confirmed by biopsy) Surgery: TAD followed by ALND Marking technique: Radioactive iodine seed	Identification rate, accuracy, and false negative rate
**TATTOO** DRKS00013169	Completed, full results pending [47,66]	Single-arm trial cN+ patients (confirmed by biopsy) Surgery: TAD or TLNB + ALND Marking technique: carbon suspension	Detection rate of the TLN
**TAXIS** [67] NCT03513614	Recruiting	Randomized phase III trial cN+ patients Surgery: tailored axillary surgery with or without ALND followed by radiotherapy	DFS
**NCT03718455**	Terminated due to limited operating room availability	Single-arm trial cN+ patients (confirmed by biopsy) Marking technique: Magnetic seed	Detection rate of the TLN
**NCT03411070**	Recruiting	Single-arm trial cN+ patients Surgery: Marking technique: radar reflector-localization clip	Rate of successful removal of the TLN

Abbreviations: ALND—axillary lymph node dissection, ART—axillary radiation therapy, DFS—disease-free survival, iDFS—invasive disease-free survival, TAD—targeted axillary dissection, TLN—target lymph node.

## Data Availability

No new data were created or analyzed in this study. Data sharing is not applicable to this article.

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
