# Peer review of "Surgical Management of the Axilla in Clinically Node-Positive Breast Cancer Patients Converting to Clinical Node Negativity through Neoadjuvant Chemotherapy: Current Status, Knowledge Gaps, and Rationale for the EUBREAST-03 AXSANA Study"

_cancers, 2021, doi:10.3390/cancers13071565_

Round 1
Reviewer 1 Report
Please specify the difference between SNLB, TLNB and TAD.
In Table 8, the TAXIS-Study (Tailored axillary surgery for clinically node-positive breast cancer in the upfront surgery and neoadjuvant setting: Prospective study within SAKK 23/16, IBCSG 57-18, ABCSG-53, GBG 10) should also be mentioned, as it includes NACT patients who convert from pN1 to ypN0.
Author Response
Please specify the difference between SNLB, TLNB and TAD.
Good idea. We included a new table (Table 1) with all definitions.
In Table 8, the TAXIS-Study (Tailored axillary surgery for clinically node-positive breast cancer in the upfront surgery and neoadjuvant setting: Prospective study within SAKK 23/16, IBCSG 57-18, ABCSG-53, GBG 10) should also be mentioned, as it includes NACT patients who convert from pN1 to ypN0.
We included the TAXIS trial, thank you for mentioning this important study!
Reviewer 2 Report
I recommend that the authors highlight that the SLNB alone is associated with a FNR of 13% which is above the 105 target and cite this meta-analysis:
https://pubmed.ncbi.nlm.nih.gov/27671032/
In the discussion the authors should discuss the impact of molecular subtype on nodal pCR. Patients with TNBC or Her2+ breast cancer who achieve a complete radiological breast pCR (on MRI or USS) are most likely to achieve pCR and benefit from TAD.
Reflector-guided localization is associated with minimal MRI artefact (rather than possible artefact)
The MRI artefacts associated with Magseed prohibit its deployment at the time of biopsy
The authors should clearly defined that TAD refers to marked lymph node biopsy plus SLNB.
Author Response
I recommend that the authors highlight that the SLNB alone is associated with a FNR of 13% which is above the 105 target and cite this meta-analysis:
https://pubmed.ncbi.nlm.nih.gov/27671032/
Good point, we included the meta-analysis in Introduction and Table 3.
In the discussion the authors should discuss the impact of molecular subtype on nodal pCR. Patients with TNBC or Her2+ breast cancer who achieve a complete radiological breast pCR (on MRI or USS) are most likely to achieve pCR and benefit from TAD.
We expanded this part of discussion.
Reflector-guided localization is associated with minimal MRI artefact (rather than possible artefact)
Corrected.
The MRI artefacts associated with Magseed prohibit its deployment at the time of biopsy
We commented on this important issue.
The authors should clearly defined that TAD refers to marked lymph node biopsy plus SLNB.
We included definitions of surgical staging techniques in Table 1.
Reviewer 3 Report
This is a comprehensive review of the data. I would recommend making it more of a cohesive story. The authors report a lot of data, but do not propose any conclusions. Also, it would be interesting to see how the authors think that AXSANA will contribute. What questions do they think it will answer? Since all treatment is left to clinicians discretion, there is a risk that the groups will be too heterogeneous for data.
Author Response
This is a comprehensive review of the data. I would recommend making it more of a cohesive story. The authors report a lot of data, but do not propose any conclusions. Also, it would be interesting to see how the authors think that AXSANA will contribute. What questions do they think it will answer? Since all treatment is left to clinicians discretion, there is a risk that the groups will be too heterogeneous for data.
Reply:
Thank you for these comments. We have discussed the AXSANA study, and specifically the points addressed by the reviewer, in more detail.
Round 2
Reviewer 3 Report
OK for publication